# Flood Mapping Uncertainty from a Restoration Perspective: A Practical Case Study

**Cássio G. Rampinelli *, Ian Knack * and Tyler Smith**

Department of Civil and Environmental Engineering, Clarkson University, 8 Clarkson Avenue, Box 5710, Potsdam, NY 136991, USA; tsmith@clarkson.edu
* Correspondence: cassiorampinelli@gmail.com (C.G.R.); iknack@clarkson.edu (I.K.)

**Abstract:** Many hydrologic studies that are the basis for water resources planning and management rely on streamflow information. Calibration and use of hydrologic models to extend flow series based on rainfall data, perform flood frequency analysis, or develop flood maps for land use planning and design of engineering works, such as channels, dams, bridges, and water intake, are examples of such studies. In most real-world engineering applications, errors in flow data are neglected or not adequately addressed. However, because flows are estimated based on the water level measurements by fitted rating curves, they can be subjected to significant uncertainties. How large these uncertainties are and how they can impact the results of such studies is a topic of interest for researchers, practitioners, and decision-makers of water resources. The quantitative assessment of these uncertainties is important to obtain a more realistic description of many water resources related studies. River restoration in many areas is limited by data availability and funding. A means to assess the uncertainty of flow data to be used in the design and analysis of river restoration projects that is cost effective and has minimal data requirements would greatly improve the reliability of river restoration design. This paper proposes an assessment of how uncertainties related to rating curves and frequency analysis may affect the results of flood mapping in a real-world application to a small watershed with limited data. A Bayesian approach was performed to obtain the posterior distributions for the model parameters and the HEC-RAS (Hydrologic Engineering Center-River Analysis System) hydraulic model was used to propagate the uncertainties in the water surface elevation profiles. The analysis was conducted using freely available data and open source software, greatly reducing traditional analysis costs. The results demonstrate that for the study case the uncertainty related to the frequency analysis study impacted the water profiles more significantly than the uncertainty associated with the rating curve.

**Keywords:** flood mapping; uncertainty; Bayesian inference; rating curve

## 1. Introduction

River restoration is a prominent area of applied water resources science and involves a variety of modifications in rivers ecosystems and stream riparian zones embracing different purposes to improve hydrologic, geomorphic, and/or ecological processes in degraded watersheds [1,2]. Examples of such overarching purposes include aesthetics, recreation, education, bank stabilization, channel reconfiguration, fish passage, floodplain reconnection, flow modification, land acquisition, instream habitat, and species improvements and management [1]. Regardless of the river restoration goals, the essence behind such initiatives is that restoring rivers to a more natural status is important not only for purely environmental reasons but also to reduce flood and geomorphic risks, besides reducing or avoiding costs of operation, maintenance, and replacement of hard works interventions [3]. Often, these restoration projects are located in areas with limited data resources and have limited

financial resources. These limitations greatly reduce the ability to assess the uncertainties in available flow data. A cost-effective means to assess the uncertainty of flow data would help reduce over designing restoration projects to accommodate uncertainties.

In this context, many multi-purpose restoration plans are triggered by flooding and related natural disasters that have a significant impact on socio-economic activities of populations in developed and developing countries around the world [4,5]. Thus, the process of determining inundation extents by the development of flood hazard maps, including the frequency of floods, and how they affect infrastructure and social activities in flood-prone areas is part of any restoration plan. The procedure generally runs through estimating the magnitude of a flood with a particular likelihood, simulating that flood in a hydraulic model, and delineating the resulting flood extents, which are represented as a deterministic boundary [6]. Deterministic flood hazard boundary delineations tend to induce the use of the resulting information for human development patterns closely following them [7,8]. This has revealed shortcomings as a result of neglecting uncertainty when delineating flood boundaries, since flood insurance claims outside regulatory flood hazard boundaries have occurred more frequently [8–11]. Addressing uncertainty can not only support the decision-making process on direct considerations regarding outputs of hydraulic/hydrologic studies but also on data acquisition. Since many projects suffer from a lack of funds, the possibility of including the impact of uncertainty in model outputs allows the use of decision analyses to assess if it is worth investing time and resources to acquire additional data and/or to perform further site investigations to better constrain the uncertainty in model evaluations [12]. From a broader perspective, uncertainty analysis enhances the scientific understanding of hydrodynamic modeling in river, climate, coastal, and environmental systems. Three main steps should be considered to systematically address uncertainty: identification of the sources of uncertainty, quantification of uncertainty from different sources, and proper communication of the uncertainty impact on the study outcomes [7]. Here, our interest focuses on corroborating to the first and second stages of the uncertainty assessment.

Among the different sources of uncertainty, two basic kinds can be typified: natural and epistemic uncertainties [13]. Natural uncertainty is related to the variability of the underlying stochastic process while epistemic uncertainty results from incomplete knowledge about the study processes [13]. Another way of typifying the sources of uncertainty is by focusing on model processes and associating it to the choice of [7]: model structures [14,15], model parameters [14,16], model inputs [14,17,18], validation data [19], change in floodplain landscape over time [8], and change in climate conditions [20,21]. A more thorough definition of the different facets of uncertainty is presented in [22] in terms of aleatory, epistemic, ontological, and linguistic uncertainties.

For this study, we used a broader and more practical perspective in which the uncertainty associated with the development of flood maps is related to the uncertainties that arise from hydrologic, hydraulic, and topographic analyses. The objective of the hydrologic study is to estimate the magnitude of floods for different return periods by applying frequency analysis, regionalization methods, and/or hydrologic modeling when needed. In most real-world engineering applications, errors in flow data are neglected or not adequately addressed, and model outputs are assumed to be deterministic. However, because flows are estimated based on the water level measurements and the rating curve, it is, in reality, affected by uncertainties [23]. The rating curve is a mathematical function that relates stage measurements with discharge values at a given station. Therefore, the rating curve is only an approximation of the real relationship between water levels and discharge values which is reflected in uncertainties in the daily streamflow data. Such uncertainties can be expressed in terms of the rating curve parameters and will be reflected not only on discharge estimated for given return periods but also on the boundary conditions of the hydraulic model. In flood frequency analysis, these uncertainties can be even larger because a relatively significant portion of the data is estimated based on the extrapolation of the rating curve [24], not to mention inherent uncertainty related to the assumptions of the statistical model and hydraulic channel control or flow regime that can be affected by temporary or permanent

changes due to seasonal vegetation growth, variation of boundary conditions or hysteresis due to transient flow conditions [25].

As part of the hydraulic study the flow is propagated over the terrain by a hydraulic model. This involves the definition of the boundary conditions for the study reach, which entails setting a rating curve and water surface elevations to the boundaries of the study reach. After defining the hydraulic model to be used, friction parameters to characterize channel and floodplain roughness should also be adjusted, aggregating more uncertainty to the process. Finally, the terrain on which the flow is routed is obtained by a topographic study, which is generally the result of a digital elevation model that merges bed topography (bathymetry) and emerged terrain (topography) based on field survey data and geoprocessing.

How large these uncertainties are and how they can impact the results of the aforementioned studies have been a topic of interest for researchers, practitioners, and decision-makers in the water resources science embracing uncertainties related to boundary conditions [8,26], spatial resolution and model structure [27–31], roughness [32–34], rating curve [25], non-stationarity [35], and general probabilistic approaches to flood mapping uncertainty [36–40]. The quantitative assessment of these uncertainties is important to obtain a more realistic description of many water resources related studies. Bayesian inference is very attractive in these cases because it can easily incorporate the often imprecise knowledge available on the hydraulic behavior of the river into the flood frequency analysis, providing a natural way to not only evaluate the uncertainties in the streamflow sample but also to consider these uncertainties in the estimated flood quantiles [24].

A fully Bayesian model proposed in [24] allows the integrated estimation of the uncertainties from the rating curve and in the flood frequency analysis. Most of the reported studies generally consider such uncertainties separated either for the rating curve parameters [41–43] or for the frequency analysis [44–48]. Although there are a few studies addressing flood mapping uncertainties [6,7,49–51], there is a lack of approaches that investigate how the combined uncertainties related to the parameters from the rating curve and the statistical distributions can affect flood mapping. In this case study, given the limited data available, we focused on the propagation of these two sources of uncertainty in flood mapping, although other sources of uncertainty may also be of relevance.

Additionally, most of the reported studies focus on research applications based on extensive data to support a broad theoretical background to address the different sources of uncertainty. This academic context limits such applications to real-world cases, especially for low-budget engineering projects on small stream reaches. In this paper, we demonstrate, by presenting a practical study case on the East Branch of the Ausable River, Keenee Valley, NY, USA, that even with limited data (poor flow measurements and topography), and a constrained budget, it is possible to obtain reasonable outputs to support decision-making under uncertainty based on flood mapping. The study aimed to quantify to what extent flood maps derived from a hydraulic model can be affected by uncertainties related to the rating curve and frequency analysis as discussed in the Bayesian model proposed in [24] that is capable of combining uncertainties from the rating curve and flood frequency analysis. More specifically, we were interested in generating flood maps for different return periods including the 95% credible intervals for the flood boundaries to the return periods of 3, 100, 1000, and 10,000 years.

## 2. Materials and Methods

### 2.1. Site Location

The study area is located in the Noonmark reach of the East Branch Ausable River, at Keene Valley, Essex County, NY. The study reach is approximately 640 m long. The site performed well during hurricane Irene in August 2011, remaining geomorphically stable and recovering quickly. As such, the reach has been used as a model for the restoration of other reaches in the area. It has also been relatively heavily investigated for stream sections in the region. Despite the importance of the reach for river restoration activities in the region, there are relatively limited data, especially flow data,

available for the reach. The data collection includes a field campaign conducted by the Ausable River Association, which resulted in 4 surveyed cross-sections along the reach. Figure 1 presents a general overview of the study reach with photos from the 4 cross-sections and a bridge. There are no flow data available in the immediate vicinity of the study reach.

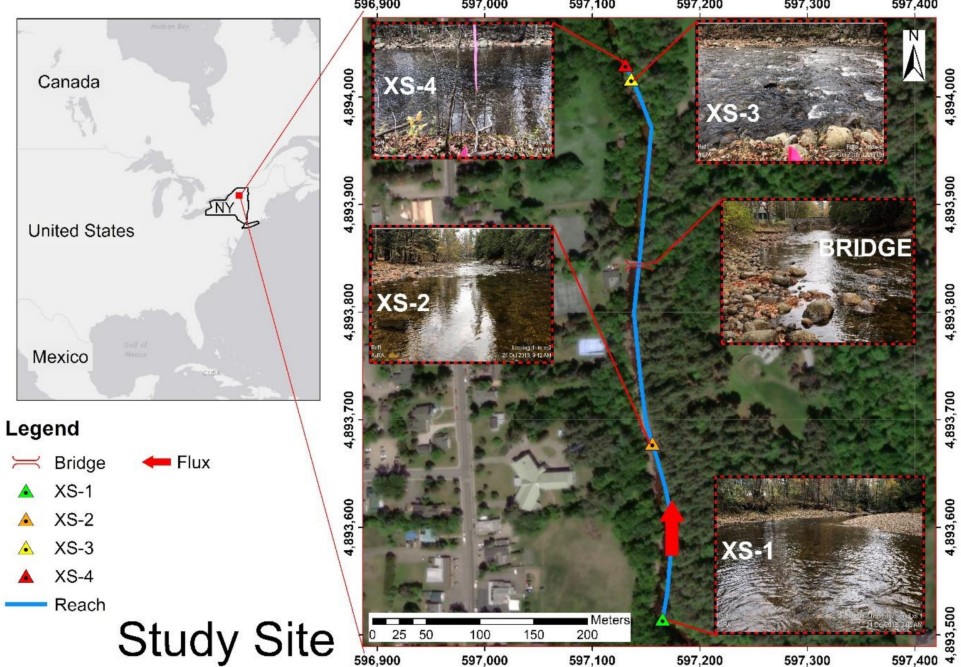

**Figure 1.** Overview of the study location.

## 2.2. Digital Elevation Model

A Digital Elevation Model (DEM) for the terrain at the site location was generated based on SRTM (Shuttle Radar Topography Mission) images. The raster images were retrieved from the National Elevation Dataset (NED) provided by the GIS New York State web site [51]. The NED is the primary data product produced and distributed by the United States Geological Survey (USGS). The NED provides seamless raster elevation data of the conterminous United States, Alaska, Hawaii, and the island territories. The NED is derived from diverse source datasets that are processed to a specification with a consistent resolution, coordinate system, elevation units, and horizontal and vertical datums. For this study, the DEM resolution was 1 m. Additional cartographic details can be accessed in [51].

The images were georeferenced and projected to the Datum D/WGS/1984, zone 18N. Some adjustments were necessary to better accommodate the contours to the topographic features. The geoprocessing tools from ArcGIS 10.2 (Esri, Redlands, CA, USA) were set to generate a Triangulated Irregular Network (TIN) for the terrain topography resulting in a Digital Elevation Model (DEM). The extension tool HEC-GeoRAS 10.2 [52] was used to processing geospatial data and to export the cross-sections delineated on the DEM to HEC-RAS 5.0.7 [53]. Extended cross-sections were delineated along the study reach, matching the 4 surveyed cross-sections. Additional cross-sections were also included in between the 4 surveyed ones. Figure 2 shows the resulting DEM, as well as the delineated cross-sections, including the stationing.

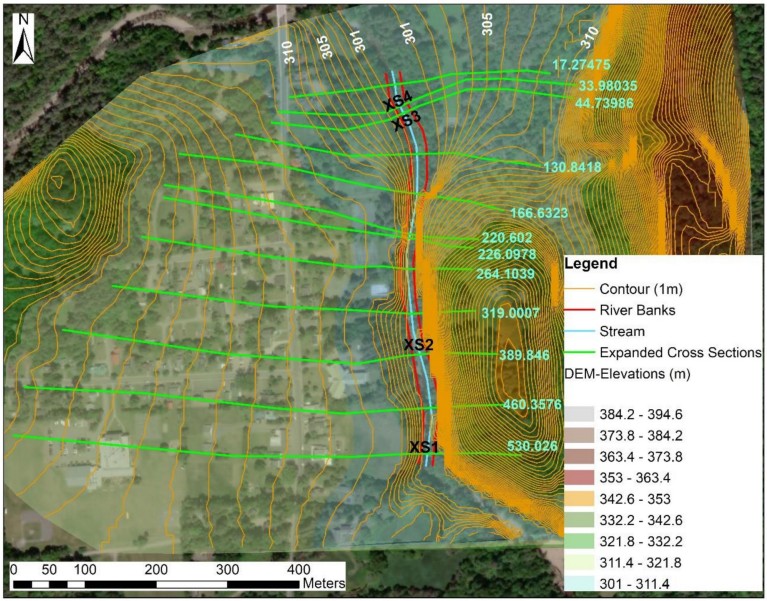

**Figure 2.** DEM and cross-sections.

### 2.3. Bathymetric Adjustments

The cross-sections derived from the DEM and those from the field survey were performed independently. Cross-sections based on the DEM were obtained from SRTM images; thus, the bathymetric portion (terrain under the water surface) was not captured by the radar. On the other hand, the cross-sections derived from the field survey, despite representing the riverbed, do not have enough length to cover the flood plain. Despite the impossibility of obtaining a perfect match between them (since there were no topographic benchmarks to do so), a comparison between the surveyed cross-sections and the DEM-retrieved cross-sections indicated the necessity of minor adjustments. These adjustments were performed by inserting the surveyed portion of the cross-section at the center of the riverbed and then merging the stations from the field survey with the cross-sections from the DEM. Additional cross-sections inserted between the surveyed ones were interpolated based on the merged (survey and DEM) cross-sections immediately upstream and downstream. Figure 3 shows an example of a comparison between a surveyed cross-section and DEM-derived cross section, as well as the resulting merged cross-section.

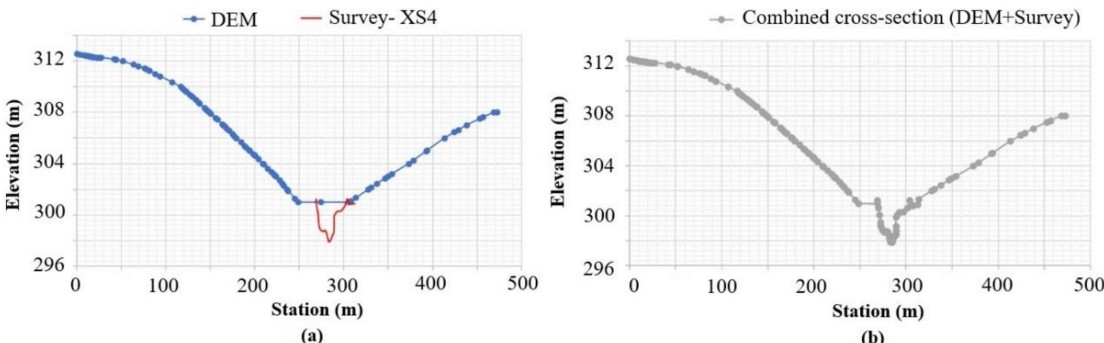

**Figure 3.** Cross-section XS4 (Station 33.98035): (**a**) comparison between field survey (continuous red line) and DEM (continuous blue line with dots); and (**b**) resulting cross-section after combining field survey and DEM data.

*2.4. Rating Curve*

2.4.1. Rating Curve Function

The rating curve represents the relationship between water level and flow. In this study, a power equation was used to fit the data. Although the method can be adjusted to represent a more complex situation in which hysteresis due to transient flow regime occurs, here, given the limited data, the flow in the river reach was assumed to be steady, and described by the following equation:

$$Q = a(h - h_0)^c \tag{1}$$

where $a$ depends on the characteristics of the river reach, $h$ is the water surface elevation, $h_0$ is the elevation associated with zero flow, and $c$ expresses the hydraulic control. The parameters were calibrated by applying the Particle Swarm Optimization algorithm [54] to minimize the Sum of Squared Errors (SSE) between flow data and simulated discharges.

2.4.2. Synthetic Stage–Discharge Data

Since no discharge–stage measurements were provided at the study location, synthetic data for discharge and stage were generated based on a hydraulic simulation performed with HEC-RAS. An interpolated cross-section positioned at station 166.6323 (refer to Figures 2 and 4) was used as a reference to the rating curve. This location was selected to avoid the influence of the downstream boundary condition at cross-section XS4 that was assumed to be the normal depth for the slope of the energy grade line (0.006 was the slope estimated based on the field survey data). For the upstream boundary condition, discharges ranging from 5 to 150 m$^3$/s were used. Based on test simulations and field data, the Manning coefficient was set to 0.034 for the channel and 0.04 for the banks. Figure 4 presents a profile of the simulated reach with an indication of the location used as a reference to derive the rating curve.

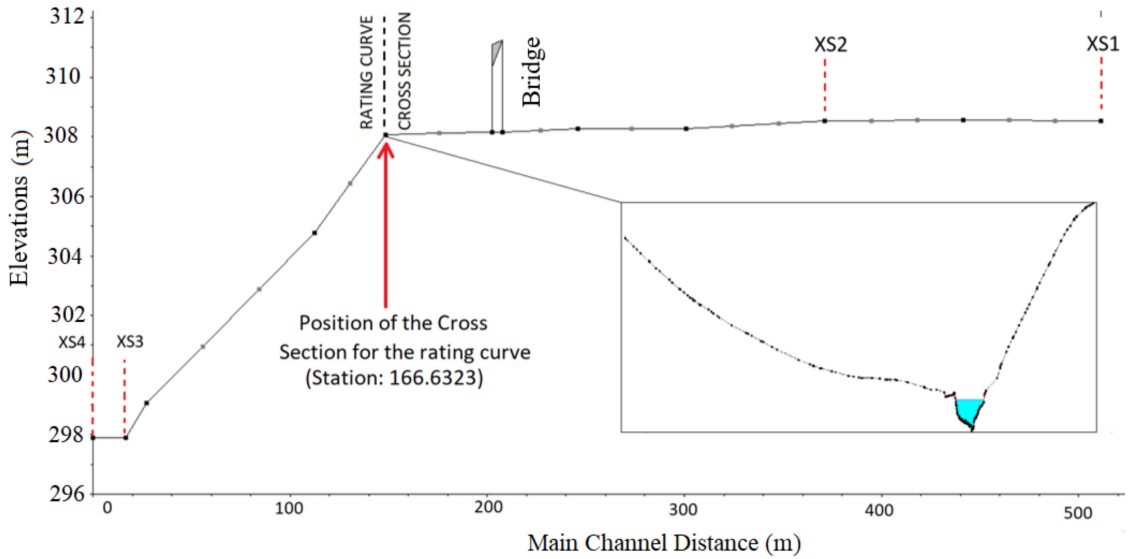

**Figure 4.** Reach profile with the indication of the surveyed cross-sections and the location of the cross-section used as reference to derive the rating curve.

*2.5. Frequency Analysis*

2.5.1. Generalized Extreme Values (GEV) Function

The three-parameter Generalized Extreme Value (GEV) distribution [55] was used for modeling the extreme discharges for the return periods 3, 100, 1000, and 10,000 years. This distribution aggregates three asymptotic forms of extreme value distributions (Frechet, Weibull, and Gumbel) in the same expression. The Cumulative Distribution Function (CDF) is given by the following expression:

$$F_X(x) = exp\left\{-\left[1 - k\left(\frac{x - \xi}{\alpha}\right)\right]^{\frac{1}{k}}\right\}; \ k \neq 0 \tag{2}$$

$$F_X(x) = exp\left\{-exp\left[-\frac{(x - \xi)}{\alpha}\right]\right\}; \ k = 0 \tag{3}$$

where κ (kappa), $\alpha$ (alpha), and $\xi$ (xi) represent the shape, scale, and position parameters, respectively. The Frechet and Weibull distributions occur for negative and positive values of κ, respectively (Equations (4) and (5)). When κ = 0, the Gumbel distribution occurs (Equation (6)), in which $x$ can assume any value.

$$Frechet \rightarrow k < 0 \rightarrow \xi + \frac{\alpha}{k} \leq x < +\infty \tag{4}$$

$$Weibull \rightarrow k > 0 \rightarrow -\infty < x \leq \xi + \frac{\alpha}{k} \tag{5}$$

$$Gumbel \rightarrow k = 0 \rightarrow -\infty < x < +\infty \tag{6}$$

The GEV Probability Density Function (PDF) is given by Equations (7) and (8):

$$f_x(x) = \frac{1}{\alpha}\left[1 - k\left(\frac{x - \xi}{\alpha}\right)\right]^{\frac{1}{k}-1} exp\left\{-\left[1 - k\left(\frac{x - \xi}{\alpha}\right)\right]^{\frac{1}{k}}\right\}; \ k \neq 0 \tag{7}$$

$$f_x(x) = \frac{1}{\alpha}exp\left\{\frac{x - \xi}{\alpha} - exp\left(\frac{x - \xi}{\alpha}\right)\right\}; \ k = 0 \tag{8}$$

The parameters from the PDF ($\alpha$, $\xi$, κ) can be estimated by the Method of Moments, Method of L-Moments, or the Maximum-Likelihood Method [55]. For this study, the method of L-Moments was used by applying the R package *lmom* [56]. Based on the estimated parameters values, the $x_p$ quantile associated with a discharge and its exceedance probability ($p$) can be computed by the following expressions.

$$x_p = \xi + \frac{\alpha}{k}\left[1 - (-\ln(p))^k\right]; \ \xi \neq 0 \tag{9}$$

$$x_p = \xi + \alpha[1 - (-\ln(p))]; \ \xi = 0 \tag{10}$$

2.5.2. Extreme Flow Data and Drainage Area Correction Factor

Since data at the study location are unavailable, the USGS flow gauge (04275000), located on the East Branch Ausable River (Latitude 44°26′14.6″, Longitude 73°40′51.5″) downstream of the study reach, was used as a reference for extreme flow data. The daily mean discharge dataset at this gauging station covers the period from 5 September 1924 to 10 September 2019, with a gap between 30 September 1995 and 8 March 2016 (Figure 5). The maximum daily mean discharge ($q_{max}$) time series at this location was used to compose the extreme value time series. The drainage area at the reference flow gauge is 512.8 km$^2$, while the drainage area at the study reach is 173.5 km$^2$. Figure 6 shows both drainage areas.

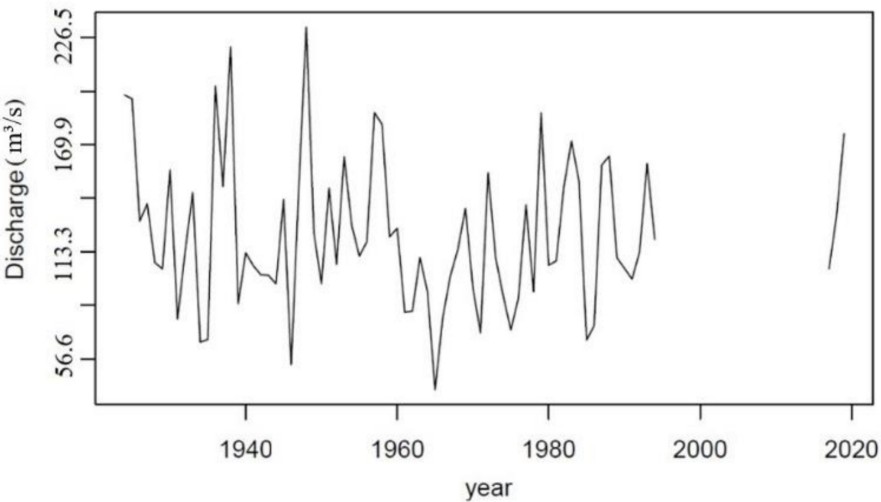

**Figure 5.** Maximum daily average discharge time series.

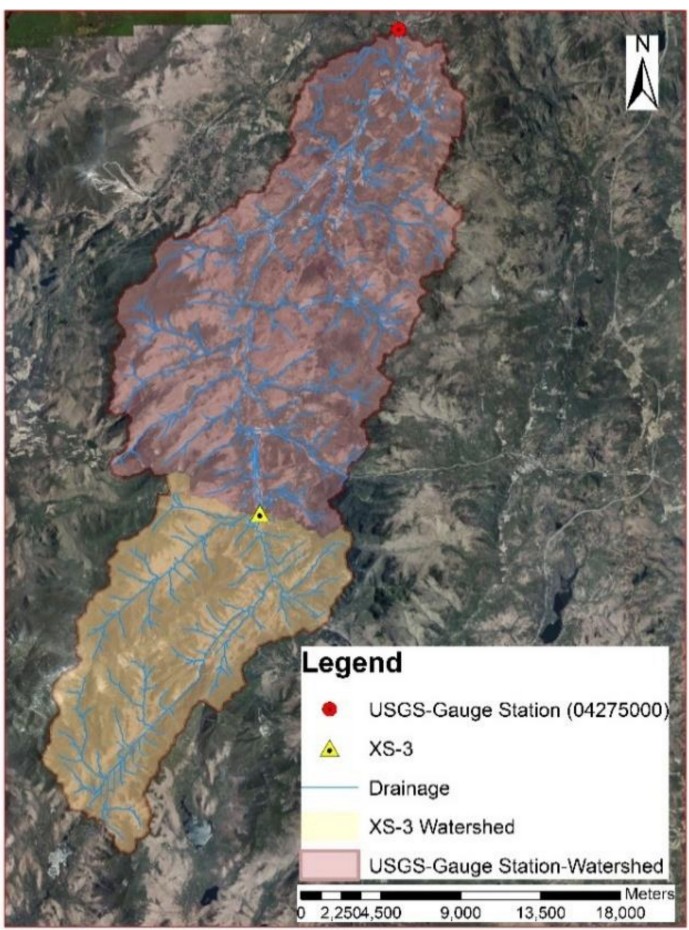

**Figure 6.** Drainage areas for the reference gauging station and the study site.

The maximum daily mean time series at the USGS gauging station was regionalized to the site location multiplying the discharges by a correction factor given by the ratio (0.338) between the drainage areas. The inverse function of the fitted rating curve was used to derive the associated stages/water elevations time series ($h_{max}$).

## 2.6. Bayesian Model

Bayesian inference is a useful approach to estimate the parameters of a given function or distribution conditioned to observed data and prior knowledge related to the model parameters. Based on Bayes theorem, the posterior distributions of the model parameters were computed considering updates on the prior knowledge by applying the likelihood function on the observed data as represented by the following expression.

$$p(\theta|x) = \frac{p(x|\theta)p(\theta)}{\int p(\theta)p(x|\theta)d\theta} \tag{11}$$

where $p(\theta|x)$ represents the posterior distribution of the parameters, $p(x|\theta)$ is the likelihood function, $p(\theta)$ is the prior distribution, and the denominator is a normalizing constant.

### 2.6.1. Bayesian Rating Curve Model

Based on [25], Bayes theorem for the relation between stage–discharge can be described by:

$$p(\theta, \sigma_f | \widetilde{H}, \widetilde{Q}) \propto p(\widetilde{Q} | \theta, \sigma_f, \widetilde{H}) p(\theta, \sigma_f) \tag{12}$$

where $p(\theta, \sigma_f | \widetilde{H}, \widetilde{Q})$ is the posterior distribution, $p(\widetilde{Q} | \theta, \sigma_f, \widetilde{H})$ is the likelihood, $p(\theta, \sigma_f)$ is the prior distribution, $\widetilde{H}$ and $\widetilde{Q}$ correspond to the measured stage–discharge pairs, $\theta$ represents the parameters of the rating curve function ($a$, $h_0$, and $c$), and $\sigma_f$ represents the standard deviation of the residuals. Since the rating curve function cannot perfectly represent the stage–discharge relation, even if the real values were known, the quality of the fit of the model is related to the magnitude of the residuals that are associated with the standard deviation.

The real discharges $\widetilde{Q}$ for each stage $\widetilde{H}$ can be represented by the sum of the discharge computed by the rating curve based on the stage–discharge function $f(\widetilde{H}_i|\theta)$ and the residuals associated with the discharge measurement and the fit of the rating curve as follows [25]:

$$\widetilde{Q}_i = f(\widetilde{H}_i|\theta) + \varepsilon_i^f + \varepsilon_i^q \tag{13}$$

$$\varepsilon_i^f + \varepsilon_i^q \sim N\left(0, \sqrt{\sigma_f^2 + u_{Q_i}^2}\right) \tag{14}$$

where $\varepsilon_i^f$ and $\varepsilon_i^q$ correspond to the fit and measurement errors, respectively. Finally, the likelihood function can be represented by the product of density functions over $N$ samples as proposed in [25].

$$p(\widetilde{Q}|\theta, \sigma_f, \widetilde{H}) = \prod_{i=1}^{N} p_N\left(\widetilde{Q}_i \middle| f(\widetilde{H}_i|\theta), \sqrt{\sigma_f^2 + u_{Q_i}^2}\right) \tag{15}$$

### 2.6.2. Fully Bayesian Model

The fully Bayesian Model presented in [24] combines rating curve and GEV parameters in an integrated posterior distribution that is proportional to the product of the likelihood function based on the observed data by the prior distributions for the parameters. The expression is presented as follows:

$$\begin{aligned} &p\left(a, h_0, c, \sigma_f, \alpha, \xi, k, u_Q \middle| \widetilde{Q}, \widetilde{H}, H_{max}, Q_{max}\right) \\ &\propto p\left(\widetilde{Q}, Q_{max} \middle| a, h_0, c, \sigma_f, \alpha, \xi, k, \widetilde{H}, H_{max}, u_Q\right) \cdot p\left(a, h_0, c, \sigma_f, \alpha, \xi, k\right) \end{aligned} \tag{16}$$

where $h_0$ and $c$ are the rating curve parameters and $\sigma_f$ represents the residuals related to the rating curve model, corresponding to the measured stage–discharge pairs $\widetilde{Q}$ and $\widetilde{H}$. $H_{max}$ and $Q_{max}$ are the time series of the maximum annual stages and associated discharges, respectively, in which the latter

depends on $H_{max}$, $a$, $h_0$, $c$, and $\sigma_f$ for each stage. Since the annual maximum discharge event $Q_{max}$ is independent of the probability of observing $\widetilde{Q}$, one can express the likelihood separated in two terms:

$$p\left(a, h_0, c, \sigma_f, \alpha, \xi, \kappa, u_Q \middle| \widetilde{Q}, \widetilde{H}, H_{max}, Q_{max}\right) \propto p\left(Q_{max} \middle| a, h_0, c, \sigma_f, \alpha, \xi, \kappa, H_{max}\right) \cdot p\left(\widetilde{Q} \middle| u_Q, a, h_0, c, \sigma_f, \widetilde{H}\right) \tag{17}$$

Assuming independent events, one can express:

$$p\left(\widetilde{Q} \middle| u_Q, a, h_0, c, \sigma_f, \widetilde{H}\right) = \prod_{i=1}^{N} p_N\left(\widetilde{Q}_i \middle| f\left(\widetilde{H}_i \middle| \theta\right), \sqrt{\sigma_{fi}^2 + u_{Q_i}^2}\right) \tag{18}$$

$$p\left(Q_{max} \middle| a, h_0, c, \sigma_f, \alpha, \xi, k, H_{max}\right) = \prod_{i=1}^{N} f_{Qmax_i}\left(q_{max_i}\right) \tag{19}$$

where

$$f_{Qmax_i}\left(q_{max_i}\right) = \frac{1}{\alpha}\left[1 - k\left(\frac{q_{max_i} - \xi}{\alpha}\right)\right]^{\frac{1}{k}-1} exp\left\{-\left[1 - k\left(\frac{q_{max_i} - \xi}{\alpha}\right)\right]^{\frac{1}{k}}\right\} \tag{20}$$

$$Qmax_i = f(H_{max} | a, h_0, c) + \varepsilon_i^f \tag{21}$$

$$\varepsilon_i^f \sim N\left(0, \sigma_f^2\right) \tag{22}$$

The standard deviation $\sigma_f$ is assumed to be heteroscedastic and linearly varying with the discharge, as follows:

$$\sigma_f = \gamma_1 + \gamma_2 Q \tag{23}$$

where $\gamma_1$ and $\gamma_2$ are parameters to be estimated.

### 2.6.3. Markov Chain Monte Carlo Simulations

To perform the Bayesian inference, a Markov Chain Monte Carlo (MCMC) algorithm was used to estimate the posterior distributions for the model parameters. The DREAM (Differential Evolution Adaptive Metropolis) algorithm originally presented in [57,58] and available as MATLAB [59] and R [60] packages was employed. Detailed information regarding the DREAM package can be found in [59].

### 2.7. HEC-RAS Model Set-Up

The HEC-RAS model [53], version 5.0.7, developed by the Hydrologic Engineering Center of the US Army Corps of Engineers was used to perform the hydraulic simulations. The model was chosen since it is freely available and extensively used for hydraulic simulations around the world. For the geometric data, as described in Section 2.2, and Section 2.3, the cross-sections were derived from a hybrid digital elevation model (DEM) resulting from a combination of SRTM images retrieved from the GIS New York State website [51], and 4 surveyed cross-sections. Twelve cross-sections were delimited, as shown in Figure 2, and exported from the DEM to the HEC-RAS with aid of the HEC-GeoRAS 10.2 [53] tool. The interpolation tool of the HEC-RAS Geometric Data module was used to create 12 more intermediate cross-sections to better represent the geometry of the reach channel. Figure 4 shows a profile plot of the study reach indicating by dark black dots the original cross-sections and by light gray dots the interpolated one.

Two sets of simulations were performed. First, simulations considering the entire profile reach were performed to provide synthetic stage–discharge data and allow the derivation of a rating curve, as described in Section 2.4. Second, simulations were performed to generate the flood maps based on four return periods, namely 3, 100, 1000, and 10,000, including the 95% credible intervals for the flood boundaries. For the upstream boundary condition, eight discharges representing the upper and lower boundaries of the 95% credible interval for the four return periods (Table 1) were used.

For the downstream boundary condition, the original profile reach (Figure 4) was trimmed at station 166.6323, where the rating curve (Figure 7) was derived and used as a downstream boundary condition. The Manning values used in the first set of simulations and presented in Section 2.4.2 were the same used for the second set of simulations, and the steady flow regime was considered. The resulting water surface profiles were exported to the ARC-GIS with aid of the HEC-GeoRAS 10.2 [52] tool where the flood maps were prepared.

**Table 1.** Mean discharges and the thresholds for 2.5% and 97.5% credibility quantiles considering different return periods (RP).

| RP | $Q$ (m$^3$/s) 2.5% | $Q$ (m$^3$/s) Med | $Q$ (m$^3$/s) 97.5% |
|---|---|---|---|
| 3 | 41.78 | 44.97 | 48.57 |
| 100 | 77 | 83.61 | 90.47 |
| 1000 | 97.22 | 106.76 | 120.98 |
| 10000 | 105.53 | 129.67 | 161.72 |

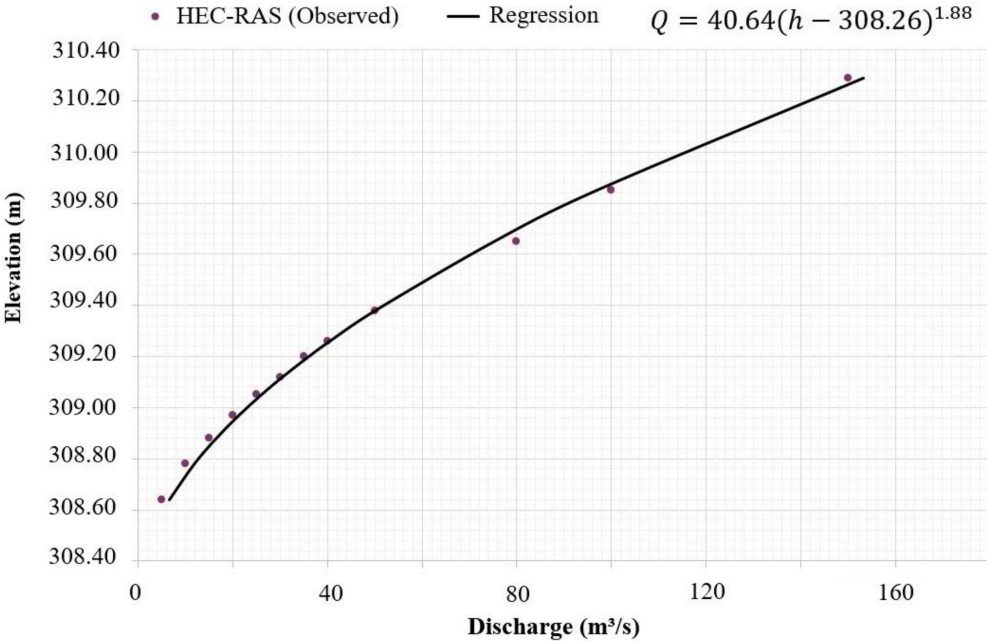

**Figure 7.** Rating curve for cross-section at station 166.6323.

## 3. Results

This section presents the results after applying the methodology described in the previous section. First, the outputs obtained for the synthetic rating curve are presented followed by the regionalized extreme data time series. Then, the results for the rating curve and the discharges resulting from the integrated Bayesian approach are shown. Finally, the flood maps are presented including the 95% credible interval.

### 3.1. Results of the Synthetic Rating Curve

After running the HEC-RAS model as described in Section 2.4.2, the water surface elevations at the rating curve reference cross-section (Figure 4) were obtained and related to the respective discharge, resulting in the synthetic observed flow-stage data used to derive the rating curve function. Based on the methodology described in Section 2.4.1, a potential function was fit to the synthetic observed data resulting in the following optimized parameters: $a = 40.64$, $h_0 = 308.26$, and $c = 1.88$. Figure 7 shows the adjusted curve as well as the synthetic data and the rating curve function, in which Q represents discharge and $h$ represents the water surface elevation.

The synthetic observed flows range from 5 to 150 m³/s while the respective elevations are between 308.64 and 310.29 m, a change of 1.65 m. Based on the field survey and the hydraulic simulations, this is considered a reasonable water level oscillation for the magnitude of the simulated discharges and should be confined to the channel banks. The range of discharges was defined to simulate a possible real-world flow measurement interval since extreme flow measurements are not usually collected. This necessitates the need to extrapolate the rating curve when performing the frequency analysis study.

## 3.2. Results of the Extreme Data Time Series at the Study Site

Before proceeding with the regionalization of the annual maximum mean daily discharges from the USGS Gauging station referred in Section 2.5.2 to the study site, the consistency of the original daily data flow was evaluated based on the flow-duration curve (Figure 8). The flow-duration curve (FDC) is the relation between the magnitudes of streamflow, q, at a point, and the frequency (probability) with which those magnitudes are exceeded over an extended period.

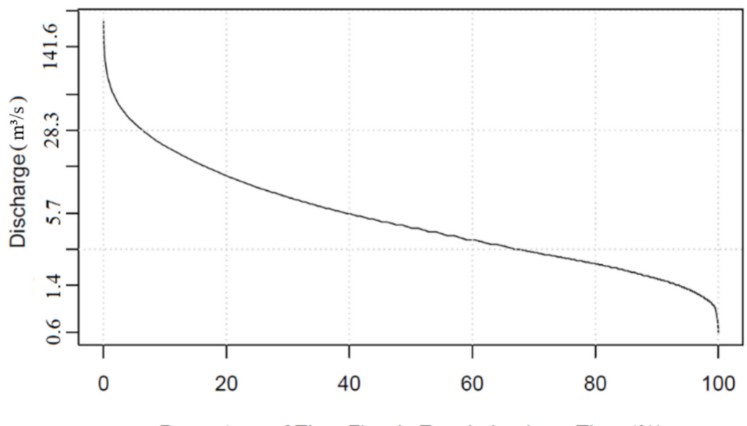

**Figure 8.** Flow duration curve at USGS flow gauge (04275000).

For the flow duration curve, the low-flow end of the curve is steep and possibly indicative of a small amount of groundwater storage above the channel bed level. Additionally, the general configuration of the obtained flow duration curve is relatively steep, representative of a variable stream. The relatively high variability of the streamflow can also be checked by comparing the discharge that is sustained 50% (4.3 m³/s) of the time with the one that is maintained 99% of the time (1.0 m³/s). The former is more than four times greater than the latter.

The annual maximum mean daily discharges at the USGS flow gauge was regionalized to the study site based on the procedure detailed in Section 2.5.2. Then, the rating curve function presented in Figure 7 was applied on annual maximum mean daily discharges time series. The adopted time interval, considered only the continuous data period from 1924 to 2019, excluding the data gap. The resulting time interval covers 74 years and is deemed reasonable to perform an applied frequency analysis.

The original and resulting extreme data time series regionalized to the study site is presented in Table A1, in Appendix A.

## 3.3. Results of the Bayesian Approach

By applying the DREAM algorithm to solve the Bayesian model for only the rating curve, as presented in Section 2.6.1, the respective 95% credible interval as well as the posteriors distributions for the rating curve parameters were obtained. The outputs are presented in Figures 9 and 10. As expected, the uncertainty increases with the magnitude of the discharge. The 95% credible interval

for parameter a varies between 40 and 58, while, for parameter $h_0$, it ranges from elevations 308.28 to 308.47 m (approximately 20 cm of variation), and, for parameter, c it varies between 1.57 and 2.

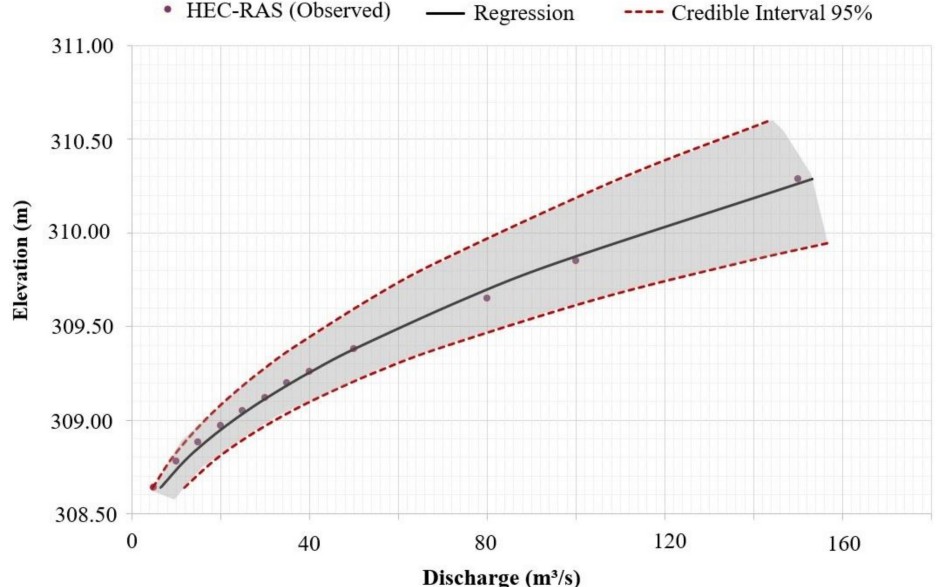

**Figure 9.** Rating curve for cross-section at station 166.6323 and 95% credible interval.

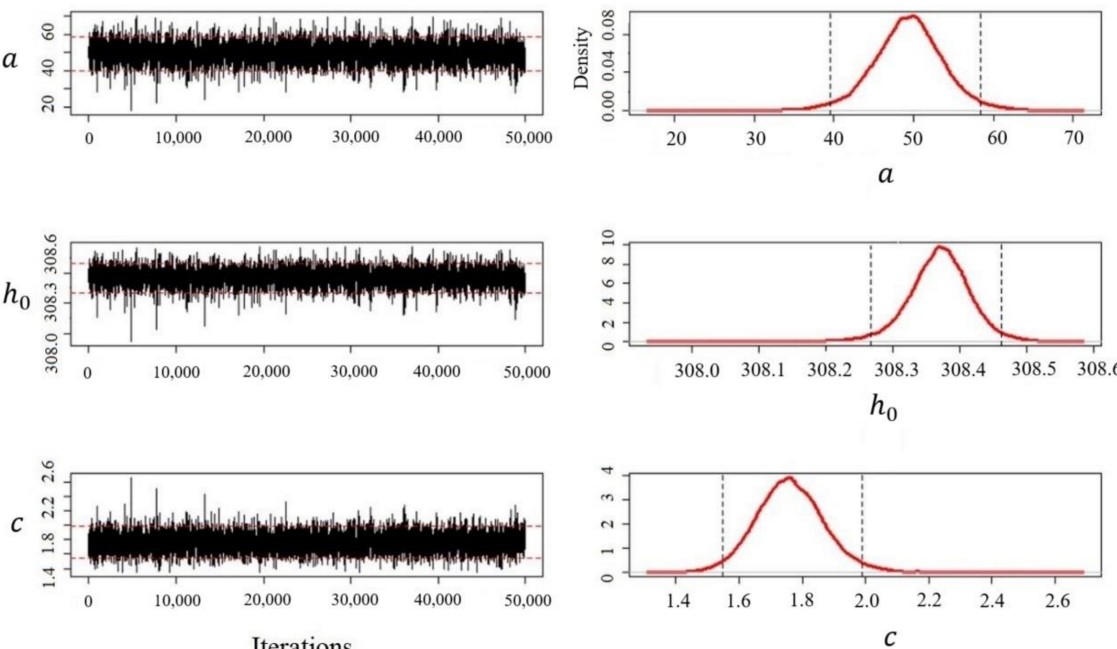

**Figure 10.** Posteriors distributions for the rating curve parameters and their 95% credible interval.

The fully Bayesian model, as described in Section 2.6.2, was applied resulting in the posterior distributions for the GEV (Figure 11) and for the extreme discharges as a function of their return periods, including the 95% credible interval (Figure 12). The 95% credible interval for the GEV parameter $\alpha$ varies between 3600 and 4700; for parameter $\xi$, it ranges from 3300 and 3430; and, for parameter $\kappa$, it varies between −0.04 and 0.06.

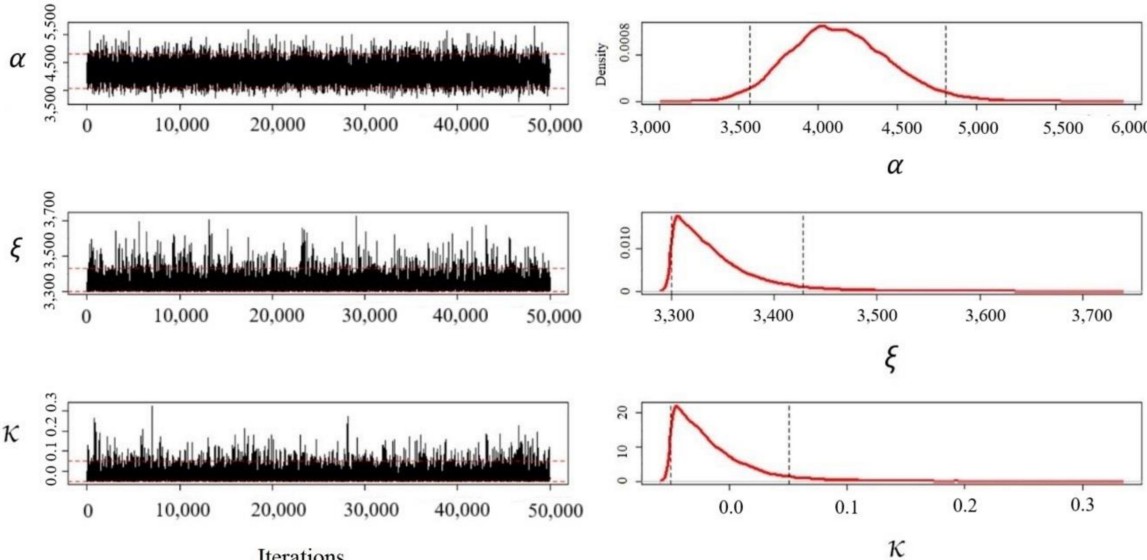

**Figure 11.** For the GEV parameters and their 95% credible interval.

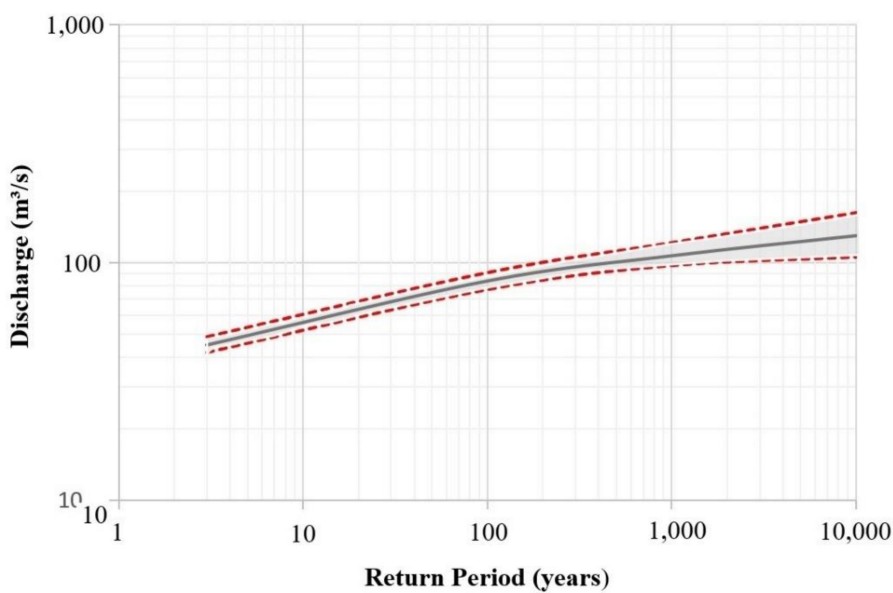

**Figure 12.** Discharges as a function of the return period and their 95% credible interval.

For the discharge as a function of the return period, as expected, as the magnitude of the flow increases so does the associated uncertainty. Table 1 summarizes the 95% credible limits for the discharge considering the return periods (RP) of 3, 100, 1000, and 10,000 years.

Figure 13 shows the uncertainty reflected on the flood maps for the return periods of 3 (Figure 13a), 100 (Figure 13b), 1000 (Figure 13c), and 10,000 (Figure 13d) years representing the upper and lower boundary of the 95% credible interval. Five buildings are indicated in Figure 13a, two of them are located near the riverbanks (b2 and b3), two of them farther away from the river (b5 and b4), and one of them in an intermediate position (b1). The impact of the simulated flood events and the value of the associated uncertainty is discussed in the next section.

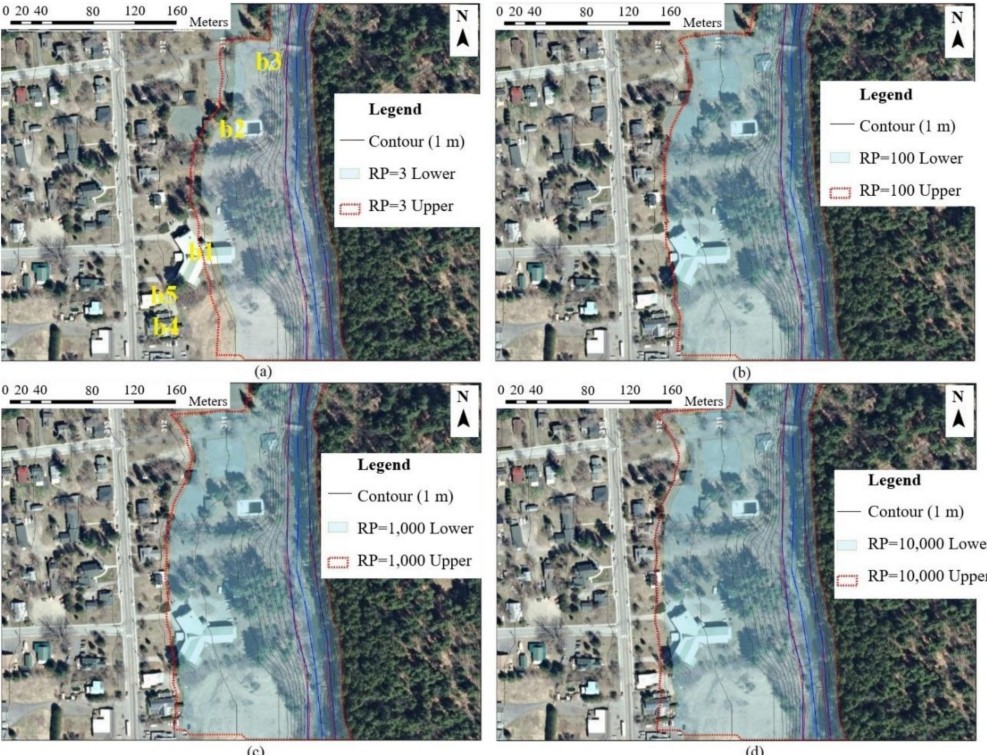

**Figure 13.** Flood maps and the 95% credible interval for the return periods: 3 years (**a**); 100 years (**b**); 1000 years (**c**); and 10,000 years (**d**).

## 4. Discussion

Despite some simplifications adopted in the model due to the lack of measured flow, the proposed methodology demonstrates the application of a framework for incorporating the uncertainty related to the rating curve and frequency analysis on the water surface profiles and flood maps to a real-world, data-limited, case study. One of the interesting aspects of the methodology applied in this study case is the fact that all the uncertainty analysis was performed using free open-source data and software. The HEC-RAS hydraulic model was employed to perform the hydraulic simulations, R was used to process the statistical and uncertainty analysis algorithms, and the topographic and hydrometric data were retrieved from free data repositories, provided by governmental agencies. This reveals the importance of open-source initiatives to the dissemination of uncertainty analysis in hydraulic and hydrologic simulations to real-world applications since a majority of studies focus on applications in which data and/or financial resources are not limiting.

The uncertainty related to the rating curve tends to increase as a function of the flow. For this study case, the maximum variation of the water level due to uncertainty was relatively low around 0.6 m (Figure 9). The low magnitude of variability is related to the fact that all data employed to derive the rating curve were artificially generated, which means that, in theory, the true values were known and correspond to the ones produced by the hydraulic model. However, it is expected that, for real measured data, the magnitude of the uncertainty could be significant.

Because the data were synthetically generated, the close fit of the regression model for the rating curve resulted in less uncertainty over the rating curve model parameters, as can be seen in Figure 10. The low degree of uncertainty refers only to the rating curve that is positioned in the control section. However, the fully Bayesian model revealed that there is much more uncertainty related to the parameters of the GEV (Figure 11) than the rating curve model parameters (Figure 10) for this case study. This is reflected in the discharge as a function of the return period (Figure 12 and Table 1) and on the generated flood maps (Figure 13). Since the uncertainty related to the roughness parameter was not considered in this study, it is expected that the consideration of such a source of uncertainty may

also affect the uncertainty boundaries and should be evaluated in future studies. Ensuring a reasonable representation of the uncertainty envelope for flooding might be of interest so that river restoration plans adjacent to the stream and riparian zone would not be unexpectedly impacted by flooding in any of the range of inundation values.

To facilitate assessment of the flood map (Figure 13), five buildings were referenced by an alphanumeric code (b1, b2, b3, b4, and b5). For practical purposes, regardless of the return period and uncertainty boundary, buildings b1, b2, and b3 can be expected to be affected. In the case of a three-year return period event (Figure 13a), building b1, near the center of the flood map (Y shaped building), would be partially affected, while a 100-year or greater event (Figure 13b–d) would be completely flood it. For these three buildings, adding uncertainty in the analysis would be indifferent to the set of actions expected from a decision-maker. Additionally, we could infer that it would not be worth investing time and financial resources to acquire more data to constrain the uncertainty limits.

On the other hand, for buildings b4 and b5, depending on the return period the uncertainty boundary starts intersecting building locations. The backyard of building b4 is intercepted by the uncertainty boundary for a 100-year return period event. For a 1000-year event, building b4 is intercepted by the 95% credible interval boundary, while the backyard of building b5 starts being affected. Finally, for the worst-case scenario, a 10,000-year return period event, both buildings are impacted by the flood uncertainty boundary. For these two buildings, in case of a deterministic flood mapping approach, they would be considered out of the flooded area. However, when uncertainty is incorporated, these buildings can be considered affected by the flood events, depending on the return period and the related uncertainty. This may be useful to consider the possibility of evaluation of the costs and value of collecting more information in constraining the uncertainty for enhanced decision making.

## 5. Conclusions

This work presents a case study to address uncertainty related to flood mapping, due to the rating curve and the frequency analysis. The quantification of the uncertainty on flood mapping for the return periods of 3, 100, 1000, and 10,000 years, including the 95% credible intervals, was performed. The Bayesian approach was applied making it possible to express the rating curve and GEV parameters as statistical distributions and addressing the impact of such uncertainty in flood mapping.

Although the uncertainty related to the rating curve did not significantly propagate through the upstream water elevation profile, in this study case, it does not mean this conclusion could be extended to any case. It is expected that regions with real measured flows should result in higher uncertainty for the rating curve. In addition, in this study case, the control section of the bridge also reduced the influence of the downstream condition on upstream cross-sections. Future studies might address such issues by including a more accurate digital elevation model, implementing in situ measuring discharge campaigns, and quantifying the uncertainty related to the topography, data regionalization, and hydraulic roughness. We highlight that acquiring more data, specifically stage–discharge measurements, is of paramount importance to validate the uncertainty analysis related to the rating curve and flow discharges. Additionally, the digital elevation model and the Manning roughness coefficient may be significant contributors to the total uncertainty.

Despite the short extension of the study reach, the data limitations, and few possible affected infrastructure, when uncertainty was incorporated into the analysis, some buildings that could not be considered affected by the flooded area in a deterministic approach became vulnerable to flood. Depending on the importance and use of the study outputs, it would be of interest to study the value of collecting more information for constraining the uncertainty and making a decision. This study demonstrates that, even in real-world case studies that suffer data limitations (both flow measurements and topography surveys), it is possible to obtain useful outputs to better inform river restoration projects and support decision-making in water resources management.

**Author Contributions:** Conceptualization, C.G.R. and I.K; methodology, C.G.R.; software, C.G.R.; validation, C.G.R.; formal analysis, C.G.R.; investigation, C.G.R.; resources, C.G.R.; data curation, C.G.R.; writing—original draft preparation, C.G.R.; writing—review and editing, C.G.R., I.K., and T.S.; visualization, C.G.R.; supervision, I.K., and T.S.; project administration, I.K. All authors have read and agreed to the published version of the manuscript.

**Funding:** This research received no external funding. However, financial support for the first author was provided by Clarkson University.

**Acknowledgments:** The authors would like to thank the Ausable River Association, Keene Valley, NY, USA for providing bathymetric and sediment data, as well as insight into the characteristics of the river reach. Additionally, the authors express their gratitude to Ana Luisa Nunes de Alencar Osorio Castañon and Prof. Dirceu Silveira Reis Jr. for sharing the R programming code that was used as a reference to support this study.

**Conflicts of Interest:** The authors declare no conflict of interest.

## Appendix A

For this table, Column 1 ($Q$ USGS) shows the annual maximum mean daily discharges time series at the USGS Gauging station; Column 2 ($Q$ Site) consists of Column 1 multiplied by the drainage factor (0.338) to regionalize the discharges to the study site; Column 3 ($H$max) shows Column 2 associated water surface elevation at the rating curve cross-section; and Column 4 (Year) is the corresponding year.

**Table A1.** Annual maximum mean daily discharges and associated stages at the rating curve cross-section.

| Q USGS (m³/s) | Q Site (m³/s) | $H_{max}$ (m) | Year | Q USGS (m³/s) | Q Site (m³/s) | $H_{max}$ (m) | Year |
|---|---|---|---|---|---|---|---|
| 196.24 | 66.40 | 309.56 | 1924 | 81.27 | 27.50 | 309.08 | 1961 |
| 194.25 | 65.73 | 309.56 | 1925 | 81.84 | 27.69 | 309.08 | 1962 |
| 129.41 | 43.79 | 309.30 | 1926 | 110.15 | 37.27 | 309.22 | 1963 |
| 138.75 | 46.95 | 309.34 | 1927 | 92.03 | 31.14 | 309.13 | 1964 |
| 107.60 | 36.41 | 309.21 | 1928 | 40.21 | 13.61 | 308.82 | 1965 |
| 104.21 | 35.26 | 309.19 | 1929 | 78.15 | 26.45 | 309.06 | 1966 |
| 156.59 | 52.99 | 309.42 | 1930 | 100.24 | 33.92 | 309.17 | 1967 |
| 77.59 | 26.25 | 309.06 | 1931 | 114.40 | 38.71 | 309.24 | 1968 |
| 113.55 | 38.42 | 309.23 | 1932 | 136.20 | 46.09 | 309.33 | 1969 |
| 144.70 | 48.96 | 309.37 | 1933 | 94.01 | 31.81 | 309.14 | 1970 |
| 65.41 | 22.13 | 308.99 | 1934 | 70.23 | 23.76 | 309.02 | 1971 |
| 66.83 | 22.61 | 309.00 | 1935 | 155.18 | 52.51 | 309.41 | 1972 |
| 201.05 | 68.03 | 309.58 | 1936 | 110.15 | 37.27 | 309.22 | 1973 |
| 147.81 | 50.02 | 309.38 | 1937 | 89.76 | 30.37 | 309.12 | 1974 |
| 221.72 | 75.03 | 309.65 | 1938 | 71.92 | 24.34 | 309.02 | 1975 |
| 85.80 | 29.03 | 309.10 | 1939 | 88.63 | 29.99 | 309.11 | 1976 |
| 112.70 | 38.14 | 309.23 | 1940 | 138.19 | 46.76 | 309.34 | 1977 |
| 105.91 | 35.84 | 309.20 | 1941 | 92.03 | 31.14 | 309.13 | 1978 |
| 101.09 | 34.21 | 309.18 | 1942 | 186.89 | 63.24 | 309.53 | 1979 |
| 100.81 | 34.11 | 309.17 | 1943 | 106.19 | 35.93 | 309.20 | 1980 |
| 96.28 | 32.58 | 309.15 | 1944 | 108.45 | 36.70 | 309.21 | 1981 |
| 141.02 | 47.72 | 309.35 | 1945 | 147.53 | 49.92 | 309.38 | 1982 |
| 53.52 | 18.11 | 308.91 | 1946 | 171.88 | 58.16 | 309.47 | 1983 |
| 146.96 | 49.73 | 309.38 | 1947 | 150.65 | 50.98 | 309.39 | 1984 |
| 232.20 | 78.57 | 309.68 | 1948 | 66.54 | 22.52 | 308.99 | 1985 |
| 123.18 | 41.68 | 309.28 | 1949 | 74.19 | 25.10 | 309.04 | 1986 |
| 96.28 | 32.58 | 309.15 | 1950 | 159.14 | 53.85 | 309.43 | 1987 |
| 146.96 | 49.73 | 309.38 | 1951 | 163.95 | 55.48 | 309.44 | 1988 |
| 106.47 | 36.03 | 309.20 | 1952 | 110.15 | 37.27 | 309.22 | 1989 |
| 163.67 | 55.38 | 309.44 | 1953 | 104.49 | 35.36 | 309.19 | 1990 |
| 127.14 | 43.02 | 309.29 | 1954 | 98.83 | 33.44 | 309.17 | 1991 |
| 111.00 | 37.56 | 309.22 | 1955 | 113.27 | 38.33 | 309.23 | 1992 |
| 118.65 | 40.15 | 309.26 | 1956 | 159.99 | 54.14 | 309.43 | 1993 |
| 186.89 | 63.24 | 309.53 | 1957 | 119.78 | 40.53 | 309.26 | 1994 |
| 180.66 | 61.13 | 309.51 | 1958 | 104.21 | 35.26 | 309.19 | 1995 |
| 121.20 | 41.01 | 309.27 | 1959 | 131.96 | 44.65 | 309.31 | 1996 |
| 125.73 | 42.54 | 309.29 | 1960 | 175.85 | 59.50 | 309.49 | 1997 |

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
