# Peer review of "Flood Mapping Uncertainty from a Restoration Perspective: A Practical Case Study"

_water, doi:10.3390/w12071948_

Round 1

Reviewer 1 Report

This manuscript presents an uncertainty analysis of rating curves and flow rate frequencies for flood mapping. Overall, I find this study interesting and practical. I only have some minor comments and suggestions for authors to consider.

Title: I am not sure how does the phrase “from a restoration perspective” relates to the topic of this study? I would suggest a more suitable title that is directly related to the topic, one possible suggestion would be “Uncertainty analysis of rating curves and frequencies for flood mapping: A practical case study” or “Quantifying uncertainty of rating curves and frequencies for flood mapping: …”, something along those lines.

Rating curves: The authors analyse and discuss uncertainties related to rating curves; however, they do not mention rating curve hysteresis. I understand that analysing the time-dependent flows is outside the scope of this study, but some discussion of the impact of hysteresis on the flood dynamics should be mentioned.

Units: I suggest using SI notation throughout the entire manuscript, specifically m^3/s or m^3 s^-1 instead of cms.

Line 131: What is the resolution of the DEM used in this study?

Methods: A subsection should be included in the Methods section describing the hydraulic (HEC-RAS) model, what type of simulation was used (steady?). What was the grid size? How were the boundaries defined? Etc…

Line 202: Why analyse 1000- and 10000-year frequencies? Do these frequencies have any practical use in the US? Considering a 74-year data set, and the purpose of the study (flood maps), I feel that frequencies up to 500 years are adequate.

Figure 10 and 11: Increase the font for the axis labels.

Lines 375-377: Figure 13 is just mentioned, some description is needed here, what do these figures show, what are the values, what are the differences? Could the uncertainty for the flood extent or flood depth at a certain point be evaluated similarly to flow rates?

Figure 12: I would suggest plotting x axis in the log scale to show the results for 3- and 100-year frequencies.

Discussion: Although the discussion adequately elaborates on the applied methodology, I feel that some comparison to similar studies is missing here.

Author Response

We thank the reviewer for the constructive comments and suggestions. We have addressed each point, individually, in the attached file below. The lines indication refer to the file water-835119_with_Review_Control.docx.

Reviewer 2 Report

It's an interesting manuscript on how uncertainties related to rating curves and frequency analysis may affect the results of flood mapping.

The manuscript is readable and correctly organized as required by quality scientific literature

The results may be of scientific interest as well as technical perspective. It is a pity that it has been applied to a small catchment, but as preliminary research it is certainly encouraging.

I have only small comments related to a better readability of figures 1,2,6 and 13.

Best wishes

Author Response

(The authors gave the same response as above.)

Reviewer 3 Report

The paper starts with two paragraphs about river restoration, which seem completely disconnected from the abstract. Then the authors say that river restoration is many times triggered by flood mitigation aims and that “… the process of determining inundation extents by the development of flood hazard maps, including the frequency of floods, and  how they affect infrastructure and social activities in flood-prone areas is part of any restoration plan”. In this way, the authors introduce the necessity of reliable mathematical modeling results. However, any intervention on rivers can generically need the support of a mathematical model. Traditional flood control measures, dredging, retention or detention reservoirs, and other actions may require the results of reliable models. It is not a prerogative of river restoration and these initial paragraphs seem to be out of place. In a first moment, I supposed that they could be part of the case study to justify or illustrate the authors’ choice, but the river restoration discussion has shown to be not particularly relevant to the paper aims.

However, the core article proposal touches a very interesting matter. As the Authors say,” the possibility of including the impact of uncertainty in model outputs allows the use of decision analyses to assess if it is worth investing time and resources to acquire additional data and/or  to perform further site investigations to better constrain the uncertainty in model evaluations”. It is a significant research question and its investigation is subject matter within this journal scope. In practice, this manuscript proposes to identify and quantify sources of uncertainty in hydrodynamic flood modelling.

In lines 72-73 the authors say that “In most cases, input flow data are considered to be free of error.” This consideration is not true, in general. Almost all modelers know that input data is full of imprecisions and that modeling is a process in which uncertainties co-exist all the time. That is why any model, to be representative, needs a calibration and validation process.

In the following, the Authors explicitly talk about rating curves uncertainties. The rating curve concept supposes a univocal relation between stages and discharges, what is really an exception in nature, being found in a few places where critical flow occurs.

In the line 93, there are 18 lumped references. This is something absolutely not informative. Please divide these references, at least, in groups and individualize the respective contributions to discussion.

At the end of the introduction, the Authors limit their research saying that “The study aims to  quantify to what extent flood maps derived from a hydraulic model can be affected by uncertainties related to the rating curve and frequency analysis”. Therefore, although opening the discussion with several possible uncertainties, the work is reduced to only two types of uncertainties.

In the line 114, what is the model proposed in [24]? The sentence would become clearer if the model is cited in an explicit way.

I do not think that the “Material and Methods” Section should initiate with the case study description. If I still do not know what will be done, that is, if the method is not already presented, I do not need to know where the method will be applied.

In the line 134, the acronym USGS appears the first time, without being presented

The mathematical proposal is tested in a case without real data. This is something fragile. In mathematical modeling, measured data is needed to calibrate the model. The obtained rating curve, as the Authors stress, was artificially generated, limiting uncertainty and weakening the paper argument about mapping uncertainties.

When looking to figure 12, the curve obtained for “discharges as a function of the return periods” is difficult to be accepted as probable. Discharges for return periods greater than 1000 years vary very little and figure 13 shows a set of simulations where the flooded areas simulated for 100, 1000 and 10000 years of return period are almost the same.

In the Conclusions, the Authors highlight that “Ensuring a reasonable representation of the uncertainty envelope for flooding might be of interest so that river restoration plans adjacent to the stream and riparian zone won’t be unexpectedly impacted by flooding in any of the range of inundation values.” They also mark five houses and assess how they could be affected when considering a range of inundation due to the calculated uncertainty, instead of a single reference return period calculation. This is an interesting discussion, but knowing that modeling brings uncertainties (greater than just the two aspects considered) an experienced modeler, when using hydrodynamic mathematical simulations to support a design decision process, would probably consider a reference return period and check the project for a greater return period, producing a flooding range, with similar results and less complication in relation to the main results produced in this manuscript.

In general, I think this manuscript should not be published in the present form. The work has potential, but it should be improved (probably, this work would benefit from further development steps).

Author Response

(The authors gave the same response as above.)

Round 2

Reviewer 3 Report

The manuscript has improved, but I still think that there are some minor issues that could be addressed.

As I said before, the main contribution of this article touches an important matter, aiming to include the impact of uncertainty in the hydrodynamic modeling process and consequent outputs. This is something important in the modeling activities, in general, but the authors introduce a bias towards river restoration demands. The Authors also say that there is an explicit Editor’s comment to keep such connections to meet the scope of a special issue. When I first reviewed this manuscript, I was not aware of it being part of a special issue.

In this context, and understanding the necessity to make the connection with river restoration discussion, I think that the authors could increase the introductory section, just to highlight that although focusing in supporting river restoration actions, this approach can be useful in a more general way, involving several uses of hydrodynamic modeling.

Another point that could be improved, in the end of the “Introduction”, is the explanation about why the manuscript is focusing on two of the uncertainties that compose the broad set of diverse uncertainties (rating curve and frequency analysis). It is important to explain the motivations and the limitations of the research. I have already made this comment previously, but I think I was not understood. The authors replied my previous question, trying to justify the wider description of the various sources of uncertainties in the introduction. However, I agree with the authors when they pose that “providing the reader with a broad perspective about the several sources and approaches regarding uncertainty is of paramount importance to indicate the limitations and the context of our study”. My intention was not to tighten the discussion, but to more clearly explain the specific choices made.

At last, I think that the conclusions section could be expanded to discuss a bit more about the limitations of the research and the future steps. These questions are only slightly cited in the present version of the text.

Author Response

We thank the reviewer for the constructive comments and suggestions. We have addressed each point, individually. Please, see the attachment.
